# Acceptance of communication technology, emotional support and subjective well-being for Chinese older adults living alone during COVID-19: A moderated mediation model

Ze Ling Nai[1]*, Woan Shin Tan[1,2], William Tov[3]

**1** Geriatric Education & Research Institute, Singapore, Singapore, **2** Health Services & Outcomes Research Department, National Healthcare Group, Singapore, Singapore, **3** School of Social Sciences, Singapore Management University, Singapore, Singapore

* nai.ze.ling@geri.com.sg

**Data Availability Statement:** Access restrictions apply to the data underlying the findings due to national data protection laws and restrictions imposed by the ethics board to ensure data privacy

## Abstract

Stringent social distancing measures implemented to control the spread of COVID-19 affected older adults living alone by limiting their social interaction beyond their households. During these restrictions, interactions beyond the household could be facilitated by communication technology (CT) such as voice calls, instant messages. Our study provides evidence on how CT acceptance could influence the emotional support and in turn, subjective well-being (SWB) of older adults living alone. We did a cross-sectional survey with 293 community-dwelling Chinese older adults. Participants were surveyed from September to November 2020 and had completed measures on CT acceptance (competency), emotional support, and SWB. PROCESS Model 1 was used to estimate the conditional effects of CT acceptance (competency) on emotional support for those living alone versus with others. Following which, PROCESS Model 7 was used to estimate the conditional indirect effects of CT acceptance (competency) on SWB through emotional support. Our results suggested that living arrangement moderated the indirect effect of CT acceptance (competency) on SWB. For older adults living alone, CT acceptance (competency) was significantly associated with perceived emotional support and, in turn, their SWB. For older adults living with others, CT acceptance was not associated with emotional support and SWB. Our findings call for more research and support to increase older adults' acceptance of CT as an option for communication to increase emotional support for older adults living alone, even during non-pandemic times.

## Introduction

In response to COVID-19, over 100 countries implemented either full or partial lockdowns with stringent social distancing measures to control the rapid transmission of the virus [1]. Some measures include limiting inter-household face-to-face interactions and maintaining

of study participants. However, anonymous minimal dataset underlying the findings in the manuscript are available upon request after authorisation by the National Health Group Domain Specific Review Board. Interested researchers may contact the Geriatric Education and Research Institute Department of Policy and Governance at enquiry@geri.com.sg.

**Funding:** Funding: This research was supported by Geriatric Education and Research Institution Limited https://www.geri.com.sg/ under its Intramural Research Grant (CG006). The funders had no role in study design, data collection and analysis, decision to publish, or preparation of the manuscript.

**Competing interests:** The authors have declared that no competing interests exist.

6-feet from others. Older adults were strongly encouraged to adhere to these safe distancing measures as they had a higher risk of developing complications from coronavirus infection [2]. These measures might have led older adults who live alone to experience greater isolation during the pandemic than older adults living with others [3], as they face more barriers to social engagement [4].

Older adults living alone were more likely to report more depressive symptoms, perceived stress and emotional distress [4–7], due to the increase barriers to social engagement. In the long term, being socially isolated can potentially lead to chronic loneliness and depression [for a review see 8]. Social engagement is an effective [9] and recommended way to mitigate the effects of social isolation [8]. However, stringent safe distancing measures could have made it difficult to maintain face-to-face social communication outside one's household. As a result, communication technology (CT; e.g. phone calls, text messages, video calls) became an essential option for everyone, including older adults to maintain social communication with others.

## Effectiveness of using CT for emotional support and SWB

Past findings suggest that CT use can alleviate older adults' social isolation by enabling connection with the outside world, gaining emotional support, and boosting self-confidence [10]. In addition, higher CT use was also associated with better self-rated health, higher SWB and fewer depressive symptoms through reduced levels of loneliness [11, 12]. Finally, CT use via text-messaging has also been found to be as effective in providing emotional support as in-person interaction with close relational partners [13] during periods of stress.

During the early stages of the pandemic, CT use also proved effective in maintaining social communication when stringent safe distancing measures were in place. Reports from China suggest that using CT (e.g. via the internet) empowered older adults to maintain intergenerational contact, which also had a direct association with their overall SWB [14]. In a mixed-methods survey in San Francisco during shelter-in-place orders, participants who experienced less loneliness reported successful use of CT (e.g. telephone, video calls and internet) for social support [15]. Similar results were found in Lombardy, where participants who used social networking sites for communication during COVID-19 reported reduced loneliness and less reduction in social contacts [16]. The use of CT (e.g. phone calls and email) for social communication also buffered the risk of higher depressive symptoms [17], especially amongst older adults living alone [6]. Using video calls to maintain contact with friends also resulted in great satisfaction in the quality of the calls, and displayed overall psychological benefits [18]. A scoping review studying the relationship between internet use and mental health in older adults during the COVID-19 pandemic suggest that internet use for communication purposes is associated with better mental health [19]. Given its implication for CT use, it is essential to understand the role of CT acceptance in assisting older adults to maintain their emotional support and SWB—especially those living alone.

We examined acceptance of CT and its implications for emotional support and SWB in a sample of Chinese older adult Singaporeans shortly after the country implemented a lockdown, known locally as the *circuit breaker* (CB). From April to June 2020, Singapore implemented nationwide stringent safe distancing measures, including limitations on inter-household visitations, social gatherings and dining out. Singaporeans were strongly encouraged to adhere to the restrictions, especially older adults who were at increased risk of serious illness and mortality due to COVID-19 [2]. During this period, older adults living alone might have depended more on social partners outside their household for support [4]. In contrast, older adults living with others had the option to seek comfort and engagement within their households. Restrictions on inter-household visitations meant that communications beyond

the household became more dependent on CT. However, older adults' use of CT is driven by their acceptance of CT [20] and confidence to use CT or CT competency [21].

Acceptance of CT, suggests that the intention to use CT is influenced by a persons' attitudes towards technology, perceived usefulness and ease of use [20]. Meanwhile, CT competency, or confidence in using CT is directly affected by a person's perceived self-efficacy of using CT, the amount of anxiety experienced, the amount of social support received and available resources to use CT. The intention to use CT is influenced by both domains [22, 23]. For example, highlighting the usefulness and providing step-by-step instructions for using CT to socialise can improve perceived usefulness, attitudes towards technology [24] and perceived ease of use [25]. Older adults who express greater enthusiasm towards technology also report greater willingness to learn, and better mental health [26]. Meanwhile, overcoming language barriers (e.g. teaching in mother tongue/dialect) and providing manuals or videos during older adults' learning journey of CT can improve their self-efficacy, and anxiety [25] when using technology. At the same time, ensuring support from family and friends and providing resources can also contribute self-efficacy when using CT [27].

The current study examines CT acceptance and competency among older adults living alone (vs not living alone) and tests whether it is associated with higher levels of SWB through perceived emotional support (Fig 1). Following Diener and colleagues [28], we examined three components of SWB: life satisfaction, positive emotion, and negative emotion. We hypothesise that for older adults living alone, acceptance of CT is likely to have a greater effect on perceptions of emotional support, which will, in turn, relate positively to their SWB. We do not expect similar effects for older adults who live with others, as the presence of other household members offers more opportunities for emotional support and social connection [4].

This study aims to close two main research gaps. Other studies have examined the relationship between CT acceptance and competency on CT use [18], as well as CT use and emotional

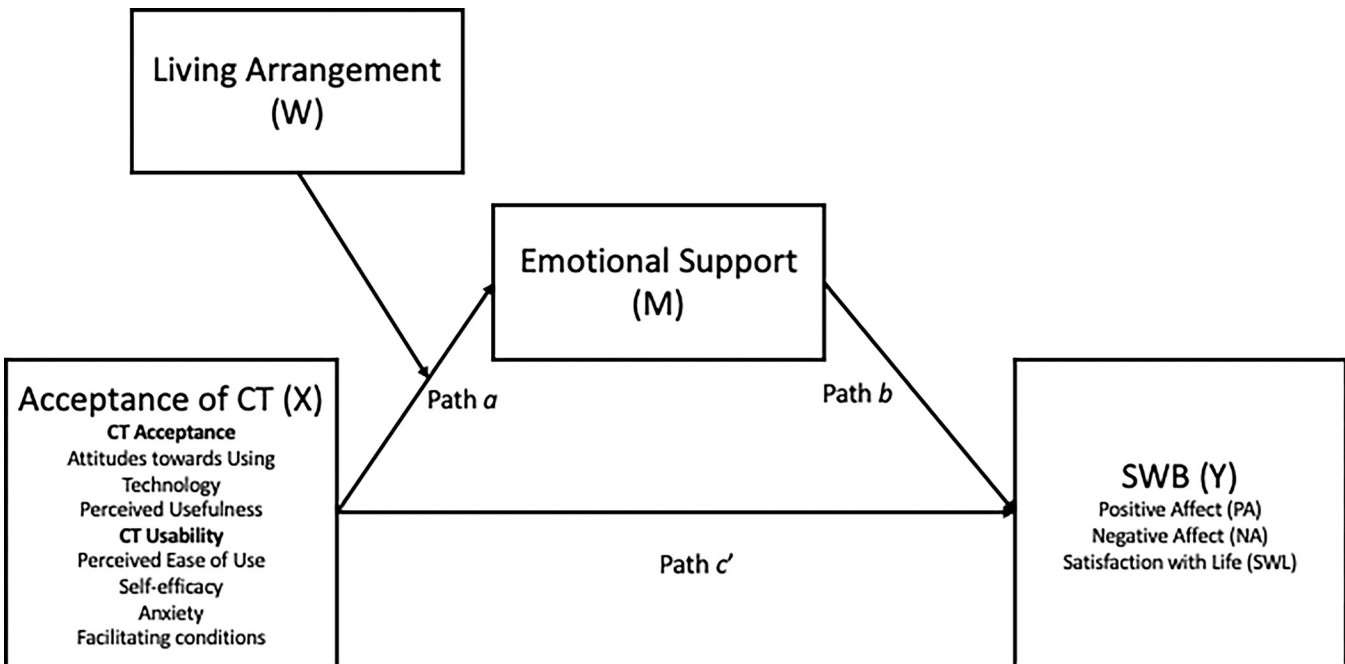

**Fig 1. Moderated mediation model between acceptance and competency of CT (X) on SWB (Y) through emotional support (M), moderated by living arrangement (W).**

support and SWB. Much of this research focused on older adults without examining the mediating effect of living arrangements. Hence, this study aims to assess the association between CT acceptance (competency) and CT use specifically amongst older adults living alone. Additionally, there is little evidence in the current literature linking CT acceptance and competency with emotional support and SWB. Hence the study also aims to determine the association between CT acceptance (competency) and their roles in providing emotional support and SWB amongst older adults living alone. Both pieces of information will be important to provide support to community care organisations and policymakers on how to address resistance and inertia amongst older adults living alone beyond the pandemic period and their daily lives. The information will also be useful in designing interventions to encourage older adults to use CT.

## Materials and methods

The study has been approved by the National Healthcare Group (NHG) Domain Specific Review Board (DSRB), with DSRB approval number 2020/00945. In line with our ethics application, written consent was taken from all participants before data collection started.

The study used a cross-sectional study design. Data collection occurred between September to November 2020 via interview-administered surveys. Participants (N = 293; $M_{age}$ = 73.8, SD = 6.5) were Chinese community-dwelling older adults. For recruitment, we partnered with four Senior Activity Centres (SACs) in Singapore's North, East and Central areas. In Singapore, SACs are found in the housing areas of communities and act as communal spaces for older adults to socialise with their peers [29]. These centres often organise activities (e.g. arts and crafts, exercise, and karaoke) and encourage older adults to participate in active ageing. The SACs publicised the study amongst their clients by word-of-mouth and posters. Interested parties were referred by the SAC managers to be screened and recruited by the study team. All were screened to ensure they were 65 years and above and spoke either English or Mandarin (or the Chinese dialects Teochew or Hokkien). Participants with diagnosed cognitive impairment and hearing problems were excluded.

Prior to the start of the survey, written informed consent was taken from all participants. Following which, participants responded to a 10-minute interview to aid recollection of their emotions and experiences during the circuit breaker (CB). Participants were encouraged to describe as vividly and as accurately as possible to aid recollection about that period. Following the interview was a 30-minute interviewer administered survey, where trained surveyors proceeded to collect participants' background characteristics, CT acceptance and competency, emotional support and SWB measures. More details of the measures used can be found in the following section.

### Measures

**Background characteristics.** We collected participants' age, education level (*no formal education*, *primary school*, *secondary school and above*), living accommodation (*studio or single bedroom apartment*, *two-bedroom apartment*, *three-bedroom apartment and above*, *condominium/landed properties*) and living arrangements (*living alone or living with others*). Similar to past studies, our study used living accommodation as an assessment of socio-economic status [30].

**Communication technology acceptance.** The Gerontology Acceptance Scale [GAS; 22] was chosen for the study as its six domains—attitudes towards technology, perceived usefulness, perceived ease of use, self-efficacy, anxiety and facilitating conditions–reflected the contributing factors of both CT acceptance and CT competency well. Participants rated 16 items

from the GAS from 1 (*strongly disagree*) to 10 (*strongly agree*). As the GAS has not been validated in Singapore, we performed an exploratory factor analysis (EFA) to ensure consistency with the original factor structure. Our results did not replicate the original factor structure. Two factors were extracted with eigenvalues greater than 1.0. We removed two items as one of the items cross-loaded onto both factors with loadings above 0.4, while the other item did not load onto either factor [loading < .32; 31]. The remaining 14 items loaded onto one of the two factors. The first factor comprised five items ($\alpha$ = .95) from attitudes towards technology, perceived usefulness and perceived ease of use, which are directly associated with acceptance and intention to use CT [20]. Hence the factor was named "CT acceptance". The second factor comprised nine items ($\alpha$ = .92) from the domains of self-efficacy, anxiety and facilitating conditions, which reflect competency in using CT [21]. Hence the factor was named "CT competency". The EFA results for the analysis are available in the S1 File.

**Emotional support.** Participants were asked to identify a target 'X', who was their main source of emotional support. Following which, participants rated the extent of emotional support received from 'X' on eight items (e.g. "X said it was OK to feel the way I was feeling") from the Support in Intimate Relationships Rating Scale [SIRRS; 32] from 0 (*never*) to 4 (*almost always*). Out of the 293 participants, 42 (14.3%) refused to respond to this scale as they had insisted that they did not have a target that provided them with emotional support. Out of the 42 participants that refused to respond, 17 participants were living alone and 25 were living with others. These participants were automatically excluded from all subsequent analysis, leaving the final sample size to be 251, of which 54 were living alone and 197 were living with others. Scores on emotional support were computed by averaging responses to the eight items ($\alpha$ = .92).

**Subjective well-being.** We examined three components of SWB: positive emotion, negative emotion and life satisfaction [28].

For positive and negative emotion, participants rated 12 positive (e.g. happy) and negative (e.g. sad) emotions from the Scale of Positive and Negative Experiences [SPANE; 33] from 1 (*very rarely or never)* to 5 (*very often or always*). Scores for positive emotions (6 items; $\alpha$ = .93) and negative emotions (6 items; $\alpha$ = .94) were generated by averaging responses to the respective subscale.

For life satisfaction, participants rated five items (e.g. "I was satisfied with my life") from the Satisfaction with Life Scale [SWLS; 34] from 1 (*strongly disagree)* to 7 (*strongly agree*). All items were averaged into a single score ($\alpha$ = .87).

**Data analysis.** All analyses were conducted using SPSS Version 26.0 [35]. Descriptive measures (i.e. frequencies, percentages) were populated for sociodemographic variables (e.g. age, gender, education) for all participants, participants living alone and participants living with others. Descriptive statistics (i.e. Mean, SD) for each variable was populated for all participants, participants living alone and participants living with others. Pearson correlations between the variables were also conducted to assess unadjusted associations between variables.

The first-step conditional mediation analyses were conducted via a two-step process using both Model 1 and Model 7 of the PROCESS macro [36]. For all models, age, education level and living accommodation were included as covariates due to their potential influence on CT acceptance and competency [37]. Language use was also included as a covariate as participants mentioned it as a contributing factor during the interviews. All predictor variables were centred by the PROCESS macro automatically to reduce multicollinearity.

Model 1 estimated the significance of the conditional effects between CT acceptance (competency), and emotional support based on living arrangements. With emotional support as the outcome (Y), we tested the effects of CT acceptance and CT competence (X) independently in two different models, which also included 'living arrangement' (0 = living with others, 1 = living

alone) as moderator (W). To assess the significance of the moderating effects, we used 95% confidence intervals (CIs) based on 5,000 bootstrapped samples [38]. Simple slopes for the conditional effects were generated. Simple slopes were considered statistically significant if the corresponding 95% confidence interval did not include zero.

Following which, Model 7 was used to estimate the conditional indirect effects based on living arrangement. We ran a total of 6 models with either CT acceptance or CT competency as independent variables (X), emotional support as mediator (M), 'living arrangement' as moderator (W) and each of the three SWB outcomes (i.e. positive emotion, negative emotion, life satisfaction) as the dependent variable (Y). Significance of the model conditional indirect effects was determined by inferencing the 95% CIs based on 5,000 bootstrapped samples [38]. Indirect effects were considered statistically significant if the corresponding 95% confidence interval did not include zero.

Multiple testing was not adjusted for in this study despite the number of models tested, as correcting for the p-value might reduce the power of the study. Power for moderation effects is generally low [39], and adjusting the p-value might increase type II error.

## Results

### Descriptive statistics

Seventy-one older adults lived alone, with 222 living with others (e.g., family, friends, foreign domestic helper, or tenant). Table 1 presents the descriptive statistics and demographic variables for the current sample.

### Correlations

CT acceptance was not significantly correlated with emotional support and the various SWB measures (Table 2). On the other hand, CT competency was significantly correlated with the various SWB components (positive emotion, negative emotion, life satisfaction). Consistent with past studies, emotional support was correlated with positive emotions and life satisfaction. However, we did not find a significant correlation between emotional support and negative emotions.

### Results from moderated mediation

Reporting the results of the moderated mediation analyses start with the moderation analysis, followed by tests of the conditional indirect effects.

**Effects of CT acceptance on emotional support and SWB.** Results for the moderated mediation analyses for CT acceptance on SWB outcomes are shown in Table 3. The effect of CT acceptance on emotional support (path *a* in Fig 1) was moderated by living arrangement (*b* = .12, 95% CI[.01; 0.24]). A simple slopes analysis indicated that the effect of CT acceptance on emotional support (path *a*) was significant for older adults living alone (*b* = .13, 95% CI [.02; .23]) but not for those living with others (*b* = -.01, 95% CI[-.04; .05]). There were also significant associations (path *b* in Fig 1) between emotional support and positive emotions and life satisfaction, but not negative emotion. For older adults living alone, CT acceptance also had an indirect positive effect (path *a*\* path *b*) on positive emotion and life satisfaction, but not for negative emotion. For older adults living with others, no indirect effects of CT acceptance were observed.

**Effects of CT Competency on emotional support and SWB.** Results for the moderated mediation analyses for CT competency on SWB outcomes are shown in Table 4. The effect of CT competency on emotional support (path *a* in Fig 1) was also moderated by living

**Table 1. Demographic and variable descriptive statistics.**

| Variable | | Living alone (n = 71) | Living with others[1] (n = 222) | Total (N = 293) |
|---|---|---|---|---|
| Demographics (n, %) | | | | |
| | Mean age (SD) | 75.5 (6.6) | 73.3 (6.4) | 73.8 (6.5) |
| | Gender | | | |
| | Male | 15 (21.2%) | 72 (32.4%) | 87 (29.7%) |
| | Female | 56 (78.9%) | 150 (67.6%) | 206 (70.3%) |
| | Preferred Language | | | |
| | English | 21 (29.6%) | 59 (26.6%) | 80 (27.3%) |
| | Mandarin (Chinese) | 49 (69.0%) | 155 (69.8%) | 204 (69.6%) |
| | Others (e.g. | 1 (1.4%) | 8 (3.6%) | 9 (3.1%) |
| | Education | | | |
| | No formal education | 32 (45.1%) | 76 (34.2%) | 108 (36.9%) |
| | Primary school | 20 (28.2%) | 61 (27.5%) | 81 (27.6%) |
| | Secondary school | 16 (22.5%) | 65 (29.3%) | 81 (27.6%) |
| | Post-secondary (non-tertiary) | 1 (1.4%) | 9 (4.1%) | 10 (3.4%) |
| | Diploma, university and postgraduate | 2 (2.8%) | 11 (5.0%) | 13 (4.5%) |
| | Living accommodation | | | |
| | 1–2 room public | 36 (50.7%) | 62 (27.9%) | 98 (33.4%) |
| | 3 room public | 25 (35.2%) | 76 (34.2%) | 101 (34.5%) |
| | 4 room public | 6 (8.5%) | 53 (23.9%) | 59 (20.1%) |
| | 5 room public and private housing | 4 (5.6%) | 31 (24.0%) | 35 (11.9%) |
| **Predictors and Outcome Variables Mean (SD)** | | | | |
| Communication Technology | | | | |
| | Acceptance (SR: 1 to 10) | 7.8 (1.9) | 7.9 (2.0) | 7.9 (2.0) |
| | Competency (SR: 1 to 10) | 5.1 (1.9) | 5.6 (2.1) | 5.5 (2.0) |
| Emotional Support[2] (SR: 0 to 4) | | 2.4 (0.7) | 2.5 (0.7) | 2.5 (0.7) |
| Subjective Well-being | | | | |
| | Positive Emotion (SR: 1 to 5) | 3.6 (0.7) | 3.6 (0.7) | 3.6 (0.7) |
| | Negative Emotion (SR: 1 to 5) | 1.9 (0.9) | 1.9 (0.8) | 1.9 (0.8) |
| | Life Satisfaction (SR: 1 to 7) | 4.9 (1.2) | 5.0 (1.2) | 5.0 (1.2) |

[1]Living with others includes living with family members, friends, tenants and migrant domestic workers.

[2] Sample size for emotional support = 251, where living alone (n = 54) and living with others (n = 197).

SR = Score Range.

arrangement ($b$ = .11, 95% CI[.01; .21]). A simple slopes analysis indicated that the effect of CT competency on emotional support (path $a$) was significant for older adults living alone ($b$ = .13, 95% CI[.03; .23]) but not for those living with others ($b$ = .02, 95% CI[-.04; .07]). There were significant associations (path $b$ in Fig 1) between emotional support and positive emotions and life satisfaction, but not negative emotion. For older adults living alone, CT competency also had an indirect positive effect (path $a$* path $b$) on positive emotion and life satisfaction, but not for negative emotion. For older adults living with others, no indirect effects of CT competency were observed.

## Discussion

Stringent safe distancing measures during the pandemic may have affected older adults living alone and made it harder for them to communicate beyond their household. This study assessed older adults' acceptance and competency of CT, and their association with perception

**Table 2. Correlation between variables (N = 293).**

| Variable | | 1 | 2 | 3 | 4 | 5 | 6 |
|---|---|---|---|---|---|---|---|
| Acceptance of Communication Technology | | | | | | | |
| | 1. Communication technology acceptance | – | | | | | |
| | 2. Communication technology competency | .61** | – | | | | |
| Social support | | | | | | | |
| | 3. Emotional support[1] | .04 | .04 | – | | | |
| Subjective well-being | | | | | | | |
| | 4. Positive emotions | .08 | .16** | .27** | – | | |
| | 5. Negative emotions | -.09 | -.15* | -.11 | -.65** | – | |
| | 6. Life satisfaction | .07 | .13* | .30** | .52** | -.53** | – |

*p < .05

** p < .01

[1] Sample size for emotional support (n = 251)

**Table 3. Results of moderated mediation analysis for communication technology *acceptance* predicting subjective well-being (n = 251).**

| Moderation | | | | | | |
|---|---|---|---|---|---|---|
| Paths | | | b | SE | LLCI | ULCI |
| CT Acceptance | → | Emotional Support | -.01 | .03 | -.04 | .05 |
| Living Arrangement | → | Emotional Support | **-1.12** | **.46** | **-2.04** | **-.20** |
| CT Acceptance * Living Arrangement | → | Emotional Support | **.12** | **.06** | **.01** | **.24** |
| Simple Slopes: CT Acceptance → Emotional Support (path *a*) by Living Arrangement | | | | | | |
| | Living alone | | **.13** | **.05** | **.02** | **.23** |
| | Living with others | | -.01 | .02 | -.04 | .05 |
| Mediation Models | | | | | | |
| Paths[a] | | | b | SE | LLCI | ULCI |
| Y1: Positive Emotion | | | | | | |
| Emotional Support | → | Positive Emotion (path *b*) | **.27** | **.06** | **.14** | **.39** |
| CT Acceptance (Emotional Support) | → | Positive Emotion (path *c'*) | **.06** | **.02** | **.02** | **.11** |
| Conditional Indirect Effects | | | | | | |
| | Living alone | | **.03** | **.02** | **.00** | **.08** |
| | Living with others | | .00 | .01 | -.01 | .01 |
| Y2: Negative Emotion | | | | | | |
| Emotional Support | → | Negative Emotion (path *b*) | -.13 | .07 | -.27 | .02 |
| CT Acceptance (Emotional Support) | → | Negative Emotion (path *c'*) | **-.07** | **.03** | **-.13** | **-.03** |
| Conditional Indirect Effects | | | | | | |
| | Living Alone | | -.02 | .01 | -.05 | .00 |
| | Living with Others | | -.00 | .00 | -.01 | .01 |
| Y3: Satisfaction with Life | | | | | | |
| Emotional Support | → | Life Satisfaction (path *b*) | **.45** | **.10** | **.26** | **.64** |
| CT Acceptance (Emotional Support) | → | Life Satisfaction (path *c'*) | **.09** | **.03** | **.02** | **.15** |
| Conditional Indirect Effects | | | | | | |
| | Living Alone | | **.05** | **.03** | **.00** | **.13** |
| | Living with Others | | .00 | .01 | -.02 | .03 |

Note: b–unstandardized coefficient; living arrangement (0 = living with others; 1 = living alone). Significant effects are bolded at p < .05; All analyses include age, language, education level and living accommodation as covariates. [a] Values for path-a differ for those living alone versus with others and are taken from the simple slopes analysis of CT acceptance x living arrangement.

**Table 4. Results of moderated mediation analysis for communication technology *competency* predicting subjective well-being ((n = 251).**

| Moderation | | | | | | |
|---|---|---|---|---|---|---|
| Paths | | | b | SE | LLCI | ULCI |
| CT Competency | → | Emotional Support | .02 | .03 | -.04 | .07 |
| Living Arrangement | → | Emotional Support | **-.75** | **.30** | **-1.32** | **-.17** |
| CT Competency * Living Arrangement | → | Emotional Support | **.11** | **.05** | **.01** | **.21** |
| Simple Slopes: CT Acceptance → Emotional Support (path *a*) by Living Arrangement | | | | | | |
| | | Living alone | **.13** | **.05** | **.03** | **.22** |
| | | Living with others | .02 | .03 | -.04 | .07 |
| Mediation Models | | | | | | |
| Paths[a] | | | b | SE | LLCI | ULCI |
| Y1: Positive Emotion | | | | | | |
| Emotional Support | → | Positive Emotion (path *b*) | **.27** | **.06** | **.14** | **.39** |
| CT Competency (Emotional Support) | → | Positive Emotion (path *c'*) | **.06** | **.02** | **.02** | **.11** |
| Conditional Indirect Effects | | | | | | |
| | | Living alone | **.03** | **.01** | **.01** | **.06** |
| | | Living with others | .00 | .01 | -.01 | .02 |
| Y2: Negative Emotion | | | | | | |
| Emotional Support | → | Negative Emotion (path *b*) | -.13 | .07 | -.27 | .02 |
| CT Competency (Emotional Support) | → | Negative Emotion (path *c'*) | **-.07** | **.03** | **-.13** | **-.02** |
| Conditional Indirect Effects | | | | | | |
| | | Living Alone | -.01 | .01 | -.04 | .00 |
| | | Living with Others | -.00 | .00 | -.01 | .01 |
| Y3: Satisfaction with Life | | | | | | |
| Emotional Support | → | Life Satisfaction (path *b*) | **.44** | **.10** | **.26** | **.64** |
| CT Competency (Emotional Support) | → | Life Satisfaction (path *c'*) | **.11** | **.04** | **.04** | **.19** |
| Conditional Indirect Effects | | | | | | |
| | | Living Alone | **.05** | **.02** | **.02** | **.11** |
| | | Living with Others | .00 | .01 | -.02 | .03 |

Note: b–unstandardized coefficient; living arrangement (0 = living with others; 1 = living alone). Significant effects are bolded at p < .05; All analyses include age, language, education level and living accommodation. [a] Values for path-a differ for those living alone versus with others and are taken from the simple slopes analysis of CT competency x living arrangement.

of emotional support and SWB during the pandemic. Our findings suggest that for older adults living alone, CT acceptance and competency is associated with their SWB through emotional support. Technology allows older adults to interact socially with others beyond their household [40]. Using CT for communication reduces loneliness [11] and increases SWB [12], emphasising its value during a pandemic as a tool for social engagement amongst older adults living alone. For older adults living with others, CT acceptance and competency did not affect SWB through emotional support. The presence of people in the household may afford them more opportunities [4] and reduce dependency on CT for social interaction. Frequent in-person interaction is also associated with reduced risk of depression [41] and loneliness [42]. While COVID-19 may be treated as endemic in many countries, experts suggest that new superspreaders might emerge [43] in the future and safe distancing measures might be reintroduced. Hence, understanding how to maintain emotional support and SWB under similar circumstances is vital for the global population.

Our findings highlight the moderating effect of living arrangements on the relation between CT use and SWB. The association between CT, emotional support and SWB is prevalent in the

literature [44–46]. However, these studies usually focus on the population, and few have assessed the effectiveness of those living alone vs living with others. Older adults living alone face more obstacles for social interaction, even during non-pandemic times. They report smaller social networks, receive less instrumental and emotional support [47] and lack of companionship [48]. Hence, it is important for older adults living alone to tap on CT as a tool to effectively engage in social communication [6, 14], especially during times of stringent social distancing. Meanwhile, for older adults living with others, our findings suggests that their SWB is not dependent on CT acceptance as they have options for face-to-face contact [4, 49] with other members of the household. However, CT use can still be encouraged to maintain contact and build relationships with others beyond the household.

Practically, there is a need to consider living arrangements when prioritising the different groups of older adults to target for CT education. Older adults living alone should be a priority as there is real potential for impact on their well-being during a pandemic situation. Beyond the pandemic, the prevalence of older adults living alone will increase, and as their physical function deteriorates, they will become less community ambulant. Thus, we will need to think about how to support their emotional health and the use of CT can be a key strategy. Programmes aimed at encouraging CT use amongst older adults should explicitly consider the role of CT acceptance and competency and how these might be enhanced by enablers such as encouragement from friends and families [25, 50], and access to CT devices and data plans [23], and reducing anxiety towards CT [51].

An unexpected finding was the lack of association between emotional support and negative emotion. A potential reason could be that the relation between both emotional support and negative emotion is bi-directional and complex—particularly during a pandemic. On the one hand, emotional support might help to reduce feelings of social isolation, which are known to be associated with declines in well-being over time [52]. However, the lockdown stage of the pandemic was disruptive for many people, and this could have provoked anxiety even among those with high levels of support. One study even found that both anxiety and perceived support increased during the peak of the COVID-19 pandemic in China [53]. In Singapore specifically, although older adults living alone felt more socially isolated during the lockdown than those living with others, both groups felt more isolated and less socially satisfied than before its implementation [3]. Our relational partners can become both a source of negative emotions and the ones we seek to reduce the negative emotions [54]. Thus, although emotional support from others might generally be associated with lower levels of negative emotion, the circumstances of the pandemic could have created many anxieties for people with and without social support.

## Limitations

The study has several limitations. Firstly, we did not directly assess our participants' frequency of CT usage and social interaction despite their direct association [22]. Future studies can consider assessing CT usage and social interaction and their association with SWB. Developing an understanding of the types of CT and their frequency of use could provide information to develop targeted interventions to encourage specific types of CT amongst older adults to improve their SWB. Next, as data collection commenced a few months after the CB, participants' recollection of their experiences and emotions might be affected. The 10-minute interview was used to increase recollection accuracy. Finally, as the data are correlational, we cannot determine whether CT acceptance and competency necessarily contribute to emotional support and well-being. More generally, the use of computer technology has been found to predict well-being longitudinally [55]. However, the reverse may also be true in that people

who feel supported and have high well-being may also have more positive attitudes toward CT use. Intervention studies that introduce older adults to CT options and enhance their acceptance of and competence in using this technology would be valuable in confirming the causal role of CT beliefs and usage on well-being. Finally, 14.3% of the participants did not to respond to the emotional support scale as they had insisted during the survey that they did not have a target that provided them with emotional support. It is likely that this group of participants were a group with low social engagement, in which case, CT acceptance or competency was unlikely to affect their SWB as it was unlikely that they needed to use CT as a tool to maintain social interaction or receive emotional support. Hence, it is unlikely that the exclusion of this group would have affected our results. However, future research should aim to identify reasons for lack of social engagement to inform policies that can benefit this group of older adults. Accurate identification of these reasons could aid the respective support groups in understanding and targeting their respective needs and increase their social engagement and support from the community.

In conclusion, our findings suggest that especially for older adults who live alone, CT acceptance and competency are associated with higher levels of perceived emotional support, which in turn is associated with SWB. Future research and interventions may target older adults' CT acceptance and competency to help them overcome barriers to social interaction and improve their SWB.

## Supporting information

**S1 File. Factor analysis for GAS.**
(PDF)

## Acknowledgments

Would like to thank the participating SACs for aiding with the data collection and for referring potential participants to the study team for recruitment. We would also like to thank all the participants for taking the time during the COVID-19 period to participate in our study.

## Author Contributions

**Conceptualization:** Ze Ling Nai, Woan Shin Tan, William Tov.

**Data curation:** Ze Ling Nai.

**Formal analysis:** Ze Ling Nai.

**Funding acquisition:** Ze Ling Nai, Woan Shin Tan, William Tov.

**Investigation:** Ze Ling Nai.

**Methodology:** Ze Ling Nai, Woan Shin Tan, William Tov.

**Validation:** Ze Ling Nai.

**Writing – original draft:** Ze Ling Nai.

**Writing – review & editing:** Ze Ling Nai, Woan Shin Tan, William Tov.

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
