## [Decision Letter · Decision Letter 0]

16 Jan 2023

PONE-D-22-27277Acceptance of Communication Technology, Emotional Support and Subjective Well-being for Chinese Older Adults Living Alone during COVID-19: A Moderated Mediation ModelPLOS ONE

Dear Dr. Nai,

Thank you for submitting your manuscript to PLOS ONE. After careful consideration, we feel that it has merit but does not fully meet PLOS ONE’s publication criteria as it currently stands. Therefore, we invite you to submit a revised version of the manuscript that addresses the points raised during the review process.

We look forward to receiving your revised manuscript.

Kind regards,

Tauseef Ahmad

Academic Editor

PLOS ONE

Journal Requirements:

2. Please provide additional details regarding ethical approval in the body of your manuscript. In the Methods section, please ensure that you have specified the name of the IRB/ethics committee that approved your study.

3. Please provide additional details regarding participant consent. In the Methods section, please ensure that you have specified (1) whether consent was informed and (2) what type you obtained (for instance, written or verbal). If your study included minors, state whether you obtained consent from parents or guardians. If the need for consent was waived by the ethics committee, please include this information.

Additional Editor Comments:

In this article, there are some improvements that could be made to make it even better. The first step is to describe the gap between the current condition and the desired condition. Provide examples of how communication technology is used in conjunction with supporting data.

Reviewers' comments:

Reviewer's Responses to Questions

**Comments to the Author**

1. Is the manuscript technically sound, and do the data support the conclusions?

Reviewer #1: Partly

Reviewer #2: Yes

2. Has the statistical analysis been performed appropriately and rigorously? 

Reviewer #1: I Don't Know

Reviewer #2: Yes

3. Have the authors made all data underlying the findings in their manuscript fully available?

Reviewer #1: Yes

Reviewer #2: Yes

4. Is the manuscript presented in an intelligible fashion and written in standard English?

Reviewer #1: Yes

Reviewer #2: Yes

5. Review Comments to the Author

Reviewer #1: I'm wondering the study is relevant to current situation as it was conducted in pandemic situation in 2020. You need to validate your argument whether the findings can be useful theoretically and practically.

I also would like i suggest the method section can be more detail, specifically for the study design and how the data was analysed.

Reviewer #2: This article is good, but there are some notes for improvement to make it better. First, describe the gap between the current condition and the desired condition. Second, describe the importance of using communication technology accompanied by supporting data. Third, add journals from similar previous studies. Passion for improvement. Goodluck.

6. PLOS authors have the option to publish the peer review history of their article (what does this mean?). If published, this will include your full peer review and any attached files.

Reviewer #1: No

Reviewer #2: No

---

## [Author Response · Author response to Decision Letter 0]

2 Apr 2023

All the following information has been included in a document uploaded with the main submission titled 'Response to Reviewers'. 

Response to Editor Comments: 

We thank the editor. We have addressed each comment below carefully and in detail. We believe the quality of the manuscript has improved and sincerely hope that PLOSOne will accept it for publication. 

https://imsva91-ctp.trendmicro.com:443/wis/clicktime/v1/query?url=https%3a%2f%2fjournals.plos.org%2fplosone%2fs%2ffile%3fid%3dwjVg%2fPLOSOne%5fformatting%5fsample%5fmain%5fbody.pdf&umid=B1809329-F264-F405-A3DA-F12DF5FD7C14&auth=6e3fe59570831a389716849e93b5d483c90c3fe4-eb1b9d9024e7ff9d1ca49b2145a18dc936405229 and 

https://imsva91-ctp.trendmicro.com:443/wis/clicktime/v1/query?url=https%3a%2f%2fjournals.plos.org%2fplosone%2fs%2ffile%3fid%3dba62%2fPLOSOne%5fformatting%5fsample%5ftitle%5fauthors%5faffiliations.pdf&umid=B1809329-F264-F405-A3DA-F12DF5FD7C14&auth=6e3fe59570831a389716849e93b5d483c90c3fe4-c8caf757ed3bc401c4deea764d57e7f96d3050f0

The manuscript has been rechecked to ensure the manuscript meets PLOS ONE’s style requirements. 

2. Please provide additional details regarding ethical approval in the body of your manuscript. In the Methods section, please ensure that you have specified the name of the IRB/ethics committee that approved your study.

We have included the following paragraph stating the IRB/ethics committee that approved the study, as well as the approval number at the start of the methods section. (Page 6, line 133 to 134)

“The study has been approved by the National Healthcare Group (NHG) Domain Specific Review Board (DSRB), with DSRB approval number 2020/00945.” 

3. Please provide additional details regarding participant consent. In the Methods section, please ensure that you have specified (1) whether consent was informed and (2) what type you obtained (for instance, written or verbal). If your study included minors, state whether you obtained consent from parents or guardians. If the need for consent was waived by the ethics committee, please include this information.

Written informed consent was obtained from all participants face-to-face prior to their participation in the study. More details have been included in the methods section. The following sentence has been included in the methods section. (Page 6, line 134 to line 136)

“In line with our ethics application, written informed consent was obtained from all participants face-to-face before data collection started.”

4. We note that you have indicated that data from this study are available upon request. PLOS only allows data to be available upon request if there are legal or ethical restrictions on sharing data publicly. For more information on unacceptable data access restrictions, please see https://imsva91-ctp.trendmicro.com:443/wis/clicktime/v1/query?url=http%3a%2f%2fjournals.plos.org%2fplosone%2fs%2fdata%2davailability%23loc%2dunacceptable%2ddata%2daccess%2drestrictions&umid=B1809329-F264-F405-A3DA-F12DF5FD7C14&auth=6e3fe59570831a389716849e93b5d483c90c3fe4-4cd0e449f6768c151c56e4f4df11540bced90dd4. 

b) If there are no restrictions, please upload the minimal anonymized data set necessary to replicate your study findings as either Supporting Information files or to a stable, public repository and provide us with the relevant URLs, DOIs, or accession numbers. For a list of acceptable repositories, please see https://imsva91-ctp.trendmicro.com:443/wis/clicktime/v1/query?url=http%3a%2f%2fjournals.plos.org%2fplosone%2fs%2fdata%2davailability%23loc%2drecommended%2drepositories&umid=B1809329-F264-F405-A3DA-F12DF5FD7C14&auth=6e3fe59570831a389716849e93b5d483c90c3fe4-55f880e6bf0829d6a3ccf08331d6f4fa526c5561.

We have responded to the data availability statement in the cover letter via the following paragraph.

“With regards to our data availability, we thank you for raising this issue. Unfortunately, access restrictions apply to the data underlying the findings due to national data protection laws and restrictions imposed by the ethics board to ensure data privacy of study participants. However, anonymous minimal dataset underlying the findings in the manuscript are available upon request after authorisation by the National Health Group Domain Specific Review Board. Interested researchers may contact the Geriatric Education and Research Institute Department of Policy and Governance at enquiry@geri.com.sg.”

The full ethics statement has been included in the methods section (Page 6, line 133 to 136). The statement includes the full name of the IRB or ethics committee who had approved the study. Written informed consent was obtained from all participants prior to the start of data collection and has also been included in the statement. Please refer to questions 2 and 3 for more details. 

6. Please include captions for your Supporting Information files at the end of your manuscript, and update any in-text citations to match accordingly. Please see our Supporting Information guidelines for more information: https://imsva91-ctp.trendmicro.com:443/wis/clicktime/v1/query?url=http%3a%2f%2fjournals.plos.org%2fplosone%2fs%2fsupporting%2dinformation&umid=B1809329-F264-F405-A3DA-F12DF5FD7C14&auth=6e3fe59570831a389716849e93b5d483c90c3fe4-bfbea7a50952def9fde8cbe82f320c0d515ded70. 

Captions in the supporting information files at the end of the manuscript have been included, and the in-text citations have been matched accordingly. For this manuscript, there is only one supporting file – S1_Factor Analysis for GAS.pdf 

The citation for the supporting file has been included in the text (Page 8, line 180)

“…The EFA results for the analysis are available in S1 Appendix.” 

The following section on Supporting information has also been included at Page 26, line 554.

“S1 Appendix. Factor Analysis for GAS.”

The full reference list has been checked and updated accordingly. Changes are as follows: 

Addition of the following papers in response to reviewers’ comments regarding (1) the importance of maintaining social communication for emotional support and SWB, (2) the effectiveness of using CT for SWB and (3) implications of our findings and competency on emotional support and SWB for older adults.

5. Zaninotto P, Iob E, Demakakos P, Steptoe A. Immediate and Longer-Term Changes in the Mental Health and Well-being of Older Adults in England During the COVID-19 Pandemic. JAMA Psychiatry. 2022;79(2):151-9. doi: 10.1001/jamapsychiatry.2021.3749. (Page 3, Line 63)

6. Noguchi T, Hayashi T, Kubo Y, Tomiyama N, Ochi A, Hayashi H. Living Alone and Depressive Symptoms among Older Adults in the COVID-19 Pandemic: Role of Non–Face-to-Face Social Interactions. Journal of the American Medical Directors Association. 2023;24(1):17-21.e4. doi: https://doi.org/10.1016/j.jamda.2022.10.014. (Page 3, Line 63)

7. Savage RD, Wu W, Li J, Lawson A, Bronskill SE, Chamberlain SA, et al. Loneliness among older adults in the community during COVID-19: a cross-sectional survey in Canada. BMJ Open. 2021;11(4):e044517. doi: 10.1136/bmjopen-2020-044517.

https://doi.org/10.1186/s41256-020-00154-3 (Page 3, Line 63)

14. Li J, Zhou X. Internet use and Chinese older adults’ subjective well-being (SWB): The role of parent-child contact and relationship. Computers in Human Behavior. 2021;119:106725. doi: https://doi.org/10.1016/j.chb.2021.106725. (Page 4, Line 82)

15. Kotwal AA, Holt-Lunstad J, Newmark RL, Cenzer I, Smith AK, Covinsky KE, et al. Social Isolation and Loneliness Among San Francisco Bay Area Older Adults During the COVID-19 Shelter-in-Place Orders. J Am Geriatr Soc. 2021;69(1):20-9. Epub 20201009. doi: 10.1111/jgs.16865. PubMed PMID: 32965024; PubMed Central PMCID: PMCPMC7536935. (Page 4, Line 85)

19. Haase KR, Cosco T, Kervin L, Riadi I, O'Connell ME. Older Adults’ Experiences With Using Technology for Socialization During the COVID-19 Pandemic: Cross-sectional Survey Study. JMIR Aging. 2021;4(2):e28010. Epub 23.4.2021. doi: 10.2196/28010. PubMed PMID: 33739929. (Page 5, Line 108)

39. Rodríguez MD, Gonzalez VM, Favela J, Santana PC. Home-based communication system for older adults and their remote family. Computers in Human Behavior. 2009;25(3):609-18. doi: https://doi.org/10.1016/j.chb.2008.08.017. (Page 16, Line 323)

40. Yu RP, McCammon RJ, Ellison NB, Langa KM. The relationships that matter: social network site use and social wellbeing among older adults in the United States of America. Ageing and Society. 2015;36(9):1826-52. doi: 10.1017/s0144686x15000677. (Page 16, Line 323)

41. Wu H-Y, Chiou A-F. Social media usage, social support, intergenerational relationships, and depressive symptoms among older adults. Geriatric Nursing. 2020;41(5):615-21. doi: https://doi.org/10.1016/j.gerinurse.2020.03.016. (Page 16, Line 323)

44. Hu Y, Qian Y. COVID-19, Inter-household Contact and Mental Well-Being Among Older Adults in the US and the UK. Frontiers in Sociology. 2021;6. doi: 10.3389/fsoc.2021.714626. (Page 16, Line 331)

45. Choi NG, DiNitto DM, Marti CN, Choi BY. Telehealth Use Among Older Adults During COVID-19: Associations With Sociodemographic and Health Characteristics, Technology Device Ownership, and Technology Learning. Journal of Applied Gerontology. 2022;41(3):600-9. doi: 10.1177/07334648211047347. PubMed PMID: 34608821. (Page 16, Line 341)

46. Vroman KG, Arthanat S, Lysack C. “Who over 65 is online?” Older adults’ dispositions toward information communication technology. Computers in Human Behavior. 2015;43:156-66. doi: https://doi.org/10.1016/j.chb.2014.10.018. (Page 16, Line 342)

The following citations were removed as they were no longer applicable in the current version of manuscript: 

Mendes da Costa E, Pepersack T, Godin I, Bantuelle M, Petit B, Levêque A. Fear of falling and associated activity restriction in older people. results of a cross-sectional study conducted in a Belgian town. Arch Public Health 2012;70(1).

Taylor HO, Taylor RJ, Nguyen AW, Chatters L. Social Isolation, Depression, and Psychological Distress Among Older Adults. Journal of Aging and Health. 2018;30(2):229–46. doi: https://doi.org/10.1177/0898264316673511.

8. In this article, there are some improvements that could be made to make it even better. The first step is to describe the gap between the current condition and the desired condition. Provide examples of how communication technology is used in conjunction with supporting data.

We thank the editor for the comments. We have addressed the comments below. 

a) We have more clearly identified the research gap and have inserted the following portion in the manuscript in page 6, line 126 to 130. 

“This study aims to address two main research gaps. Firstly, while older individuals who live alone have fewer chances for social interaction (e.g. Fingerman et al., 2021), few studies have examined how CT could impact SWB across different living arrangements. Next, both acceptance and competency in using CT are highly predictive of CT use [18] and provides additional information on how to address resistance and inertia in this population” 

b) With regards to providing examples on how communication technology is used in conjunction with supporting data. We have specified the type of communication technology used in this study, and highlighted its effectiveness on improving SWB. Please refer to the insertion below from the introduction section in page 4, Line 76 to 87. 

“… CT use via text-messaging has also been found to be as effective in providing emotional support as in-person interaction with close relational partners [13] during periods of stress. 

During the early stages of the pandemic, CT use also proved effective in maintaining social communication when stringent safe distancing measures were in place. Reports from China suggest that using CT (e.g. via the internet) empowered older adults to maintain intergenerational contact, which also had a direct association with their overall SWB [14]. In a mixed-methods survey in San Francisco during shelter-in-place orders, participants who experienced less loneliness reported successful use of CT (e.g. telephone, video calls and internet) for social support [15]. It seems that the use of CT (e.g. phone calls and email) for social communication also buffered the risk of higher depressive symptoms, especially amongst older adults living alone [6].”

 

Response to Reviewers’ Comments 

Reviewer #1: I'm wondering the study is relevant to current situation as it was conducted in pandemic situation in 2020. You need to validate your argument whether the findings can be useful theoretically and practically. I also would like to suggest the method section can be more detail, specifically for the study design and how the data was analysed.

Thank you for your comments. 

1a) We have restructured the discussion section to address your concerns. Theoretically, we believe that our study contributes to the existing literature by highlighting the moderating effect of living arrangements on the relation between CT use and SWB. Practically, CT use will become increasingly important with its proliferation in different aspects of life, including our daily activities and work beyond the pandemic in 2020. There is also a need to consider living arrangement when we think about which group of older adults to prioritise and target for CT education.

We have expanded on these points in the manuscript by including the following paragraphs from Page 16, line 321 to 342. 

“Our findings highlight the moderating effect of living arrangements on the relation between CT use and SWB. The association between CT, emotional support and SWB is prevalent in the literature [39-41]. However, these studies usually focus on the population, and few have assessed the effectiveness of those living alone vs living with others. Older adults living alone face more obstacles for social interaction, even during non-pandemic times. They report smaller social networks, receive less instrumental and emotional support [42] and lack of companionship [43]. Hence, it is important for older adults living alone to tap on CT as a tool to effectively engage in social communication [6, 14], especially during times of stringent social distancing. Meanwhile, for older adults living with others, our findings suggests that their SWB is not dependent on CT acceptance as they have options for face-to-face contact [4, 44] with other members of the household. However, CT use can still be encouraged to maintain contact and build relationships with others beyond the household. 

Practically, there is a need to consider living arrangements when prioritising the different groups of older adults to target for CT education. Older adults living alone should be a priority as there is real potential for impact on their well-being during a pandemic situation. Beyond the pandemic, the prevalence of older adults living alone will increase, and as their physical function deteriorates, they will become less community ambulant. Thus, we will need to think about how to support their emotional health and the use of CT can be a key strategy. Programmes aimed at encouraging CT use amongst older adults should explicitly consider the role of CT acceptance and competency and how these might be enhanced by enablers such as encouragement from friends and families [21, 45], and access to CT devices and data plans [19], and reducing anxiety towards CT [46]. ”

1b) We have also expanded the methods section to provide more details about our study design and data analysis approach. The following portions have been incorporated into the methods section. 

Study Design: (Page 7, line 137 to 156)

“The study used a cross-sectional study design. Data collection occurred between September to November 2020 via interview-administered surveys…Interested study participants were referred by senior activity centre (SAC) managers to be screened and recruited by the study team…Prior to the start of the survey, written informed consent was taken from all participants. Following which, participants responded to a 10-minute interview to aid recollection of their emotions and experiences during the circuit breaker (CB). Participants were encouraged to describe as vividly and as accurately as possible to aid recollection about that period. Following the interview was a 30-minute interviewer administered survey, where trained surveyors proceeded to collect participants’ background characteristics, CT acceptance and competency, emotional support and SWB measures. More details of the measures can be found in the following section.”

Data Analysis: (Page 9, line 205 to 232)

“Descriptive measures (i.e. frequencies, percentages) were populated for sociodemographic variables (e.g. age, gender, education) for all participants, participants living alone and participants living with others. Descriptive statistics (i.e. Mean, SD) for each variable was populated for all participants, participants living alone and participants living with others. Pearson correlations between the variables were also conducted to assess unadjusted associations between variables. 

The first-step conditional mediation analyses were conducted via a two-step process using both Model 1 and Model 7 of the PROCESS macro [31]. For all models, age, education level and living accommodation were included as covariates due to their potential influence on CT acceptance and competency [32]. Language use was also included as a covariate as participants mentioned it as a contributing factor during the interviews. All predictor variables were centred by the PROCESS macro automatically to reduce multicollinearity. 

Model 1 estimated the significance of the conditional effects between CT acceptance (competency), and emotional support based on living arrangements. With emotional support as the outcome (Y), we tested the effects of CT acceptance and CT competence (X) independently in two different models, which also included ‘living arrangement’ (0 = living with others, 1 = living alone) as moderator (W). To assess the significance of the moderating effects, we used 95% confidence intervals (CIs) based on 5,000 bootstrapped samples [33]. Simple slopes for the conditional effects were generated. Simple slopes were considered statistically significant if the corresponding 95% confidence interval did not include zero.

Following which, Model 7 was used to estimate the conditional indirect effects based on living arrangement. We ran a total of 6 models with either CT acceptance or CT competency as independent variables (X), emotional support as mediator (M), ‘living arrangement’ as moderator (W) and each of the three SWB outcomes (i.e. positive emotion, negative emotion, life satisfaction) as the dependent variable (Y). Significance of the model conditional indirect effects was determined by inferencing the 95% CIs based on 5,000 bootstrapped samples [33]. Indirect effects were considered statistically significant if the corresponding 95% confidence interval did not include zero.” 

Reviewer #2: This article is good, but there are some notes for improvement to make it better. First, describe the gap between the current condition and the desired condition. Second, describe the importance of using communication technology accompanied by supporting data. Third, add journals from similar previous studies. Passion for improvement. Goodluck.

Thank you for your comments. 

2a) We have more clearly identified the research gap and have inserted the following portion in the manuscript in page 6, line 126 to 130

“This study aims to address two main research gaps. Firstly, while older individuals who live alone have fewer chances for social interaction (e.g. Fingerman et al., 2021), few studies have examined how CT could impact SWB across different living arrangements. Next, both acceptance and competency in using CT are highly predictive of CT use [18] and provides additional information on how to address resistance and inertia in this population.”

2b) Additional findings from the literature to address the importance of using CT supported by findings in the literature has been incorporated in two sections of the manuscript. The first section covers the impact of living arrangement on SWB, and how CT can be used as a tool to maintaining emotional support and SWB. This portion is included in the literature review. (Page 3, line 71 to line 87)

“Past findings suggest that CT use can alleviate older adults’ social isolation by enabling connection with the outside world, gaining emotional support, and boosting self-confidence [10]. In addition, higher CT use was also associated with better self-rated health, higher SWB and fewer depressive symptoms through reduced levels of loneliness [11, 12]. Finally, CT use via text-messaging has also been found to be as effective in providing emotional support as in-person interaction with close relational partners [13] during periods of stress. 

During the early stages of the pandemic, CT use also proved effective in maintaining social communication when stringent safe distancing measures were in place. Reports from China suggest that using CT (e.g. via the internet) empowered older adults to maintain intergenerational contact, which also had a direct association with their overall SWB [14]. In a mixed-methods survey in San Francisco during shelter-in-place orders, participants who experienced less loneliness reported successful use of CT (e.g. telephone, video calls and internet) for social support [15]. It seems that the use of CT (e.g. phone calls and email) for social communication also buffered the risk of higher depressive symptoms, especially amongst older adults living alone [6].” 

The second section is covered in the discussion section, where we further emphasise the use of CT not just as a tool for social communication, but its role in our daily lives in various settings such as work. (Page 16, line 321 to 342). Please refer to the response in section 1a) for more details. 

2c) In addition, we have included additional literature from similar studies. As there are few studies that have studied this topic, we have included literature into various portions of the manuscript to cover the key topics including (1) the importance of maintaining social communication for emotional support and SWB, (2) the effectiveness of using CT for SWB and (3) implications of our findings of CT acceptance and competency on emotional support and SWB for older adults. The list of the papers included in order of appearance are as shown below. 

5. Zaninotto P, Iob E, Demakakos P, Steptoe A. Immediate and Longer-Term Changes in the Mental Health and Well-being of Older Adults in England During the COVID-19 Pandemic. JAMA Psychiatry. 2022;79(2):151-9. doi: 10.1001/jamapsychiatry.2021.3749. (Page 3, Line 63)

6. Noguchi T, Hayashi T, Kubo Y, Tomiyama N, Ochi A, Hayashi H. Living Alone and Depressive Symptoms among Older Adults in the COVID-19 Pandemic: Role of Non–Face-to-Face Social Interactions. Journal of the American Medical Directors Association. 2023;24(1):17-21.e4. doi: https://doi.org/10.1016/j.jamda.2022.10.014. (Page 3, Line 63)

7. Savage RD, Wu W, Li J, Lawson A, Bronskill SE, Chamberlain SA, et al. Loneliness among older adults in the community during COVID-19: a cross-sectional survey in Canada. BMJ Open. 2021;11(4):e044517. doi: 10.1136/bmjopen-2020-044517.

https://doi.org/10.1186/s41256-020-00154-3 (Page 3, Line 63)

14. Li J, Zhou X. Internet use and Chinese older adults’ subjective well-being (SWB): The role of parent-child contact and relationship. Computers in Human Behavior. 2021;119:106725. doi: https://doi.org/10.1016/j.chb.2021.106725. (Page 4, Line 82)

15. Kotwal AA, Holt-Lunstad J, Newmark RL, Cenzer I, Smith AK, Covinsky KE, et al. Social Isolation and Loneliness Among San Francisco Bay Area Older Adults During the COVID-19 Shelter-in-Place Orders. J Am Geriatr Soc. 2021;69(1):20-9. Epub 20201009. doi: 10.1111/jgs.16865. PubMed PMID: 32965024; PubMed Central PMCID: PMCPMC7536935. (Page 4, Line 85)

19. Haase KR, Cosco T, Kervin L, Riadi I, O'Connell ME. Older Adults’ Experiences With Using Technology for Socialization During the COVID-19 Pandemic: Cross-sectional Survey Study. JMIR Aging. 2021;4(2):e28010. Epub 23.4.2021. doi: 10.2196/28010. PubMed PMID: 33739929. (Page 5, Line 108)

39. Rodríguez MD, Gonzalez VM, Favela J, Santana PC. Home-based communication system for older adults and their remote family. Computers in Human Behavior. 2009;25(3):609-18. doi: https://doi.org/10.1016/j.chb.2008.08.017. (Page 16, Line 323)

40. Yu RP, McCammon RJ, Ellison NB, Langa KM. The relationships that matter: social network site use and social wellbeing among older adults in the United States of America. Ageing and Society. 2015;36(9):1826-52. doi: 10.1017/s0144686x15000677. (Page 16, Line 323)

41. Wu H-Y, Chiou A-F. Social media usage, social support, intergenerational relationships, and depressive symptoms among older adults. Geriatric Nursing. 2020;41(5):615-21. doi: https://doi.org/10.1016/j.gerinurse.2020.03.016. (Page 16, Line 323)

44. Hu Y, Qian Y. COVID-19, Inter-household Contact and Mental Well-Being Among Older Adults in the US and the UK. Frontiers in Sociology. 2021;6. doi: 10.3389/fsoc.2021.714626. (Page 16, Line 331)

45. Choi NG, DiNitto DM, Marti CN, Choi BY. Telehealth Use Among Older Adults During COVID-19: Associations With Sociodemographic and Health Characteristics, Technology Device Ownership, and Technology Learning. Journal of Applied Gerontology. 2022;41(3):600-9. doi: 10.1177/07334648211047347. PubMed PMID: 34608821. (Page 16, Line 341)

46. Vroman KG, Arthanat S, Lysack C. “Who over 65 is online?” Older adults’ dispositions toward information communication technology. Computers in Human Behavior. 2015;43:156-66. doi: https://doi.org/10.1016/j.chb.2014.10.018. (Page 16, Line 342)

---

## [Decision Letter · Decision Letter 1]

2 May 2023

PONE-D-22-27277R1Acceptance of Communication Technology, Emotional Support and Subjective Well-being for Chinese Older Adults Living Alone during COVID-19: A Moderated Mediation ModelPLOS ONE

Dear Dr. Nai,

Thank you for submitting your manuscript to PLOS ONE. After careful consideration, we feel that it has merit but does not fully meet PLOS ONE’s publication criteria as it currently stands. Therefore, we invite you to submit a revised version of the manuscript that addresses the points raised during the review process.

We look forward to receiving your revised manuscript.

Kind regards,

Tauseef Ahmad

Academic Editor

PLOS ONE

Journal Requirements:

Additional Editor Comments:

Despite being good, this article could be improved a bit. The reviewer's comments must be corrected on some important parts of the manuscript, there are no gaps in the research, and the research methods are available.

Reviewers' comments:

Reviewer's Responses to Questions

**Comments to the Author**

1. If the authors have adequately addressed your comments raised in a previous round of review and you feel that this manuscript is now acceptable for publication, you may indicate that here to bypass the “Comments to the Author” section, enter your conflict of interest statement in the “Confidential to Editor” section, and submit your "Accept" recommendation.

Reviewer #2: All comments have been addressed

2. Is the manuscript technically sound, and do the data support the conclusions?

Reviewer #2: Yes

3. Has the statistical analysis been performed appropriately and rigorously? 

Reviewer #2: Yes

4. Have the authors made all data underlying the findings in their manuscript fully available?

Reviewer #2: Yes

5. Is the manuscript presented in an intelligible fashion and written in standard English?

Reviewer #2: Yes

6. Review Comments to the Author

Reviewer #2: This article is good, but there are some notes for improvement to make it better. First, the abstract must be corrected, where the abstract contains research methods. Second, explain in more detail along with the reasons why it is important to research your research study. Third, describe the research gap, between the current state and the desired condition, Fourth, explain what are the benefits for further research. Fifth, add journals from similar previous studies. Passion for improvement. Goodluck.

7. PLOS authors have the option to publish the peer review history of their article (what does this mean?). If published, this will include your full peer review and any attached files.

Reviewer #2: **Yes: **Rafika Bayu Kusumandari

---

## [Author Response · Author response to Decision Letter 1]

15 Jun 2023

Response to Editor Comments: 

The full reference list has been checked and updated accordingly. We have included the following papers in response to the reviewer’s comments. 

16. Rolandi, E., Vaccaro, R., Abbondanza, S., Casanova, G., Pettinato, L., Colombo, M., & Guaita, A. (2020). Loneliness and social engagement in older adults based in Lombardy during the COVID-19 lockdown: The long-term effects of a course on social networking sites use. International journal of environmental research and public health, 17(21), 7912. (Page 4, line 88)

17. Li, Y., Godai, K., Kido, M. et al. Cognitive decline and poor social relationship in older adults during COVID-19 pandemic: can information and communications technology (ICT) use helps?. BMC Geriatr 22, 375 (2022). https://doi.org/10.1186/s12877-022-03061-z (Page 4, line 89)

18. Juvonen, J., Schacter, H. L., & Lessard, L. M. (2021). Connecting electronically with friends to cope with isolation during COVID-19 pandemic. Journal of Social and Personal Relationships, 38(6), 1782-1799. (Page 4, line 92)

19. Foong, H. F., Lim, S. Y., Rokhani, F. Z., Bagat, M. F., Abdullah, S. F. Z., Hamid, T. A., & Ahmad, S. A. (2022). For better or for worse? A scoping review of the relationship between internet use and mental health in older adults during the COVID-19 pandemic. International Journal of Environmental Research and Public Health, 19(6), 3658. (Page 4, line 94)

26. Dura-Perez E, Goodman-Casanova J, Vega-Nuñez A, Guerrero-Pertiñez G, Varela-Moreno E, Garolera M, Quintana M, Cuesta-Vargas A, Barnestein-Fonseca P, Gómez Sánchez-Lafuente C, Mayoral-Cleries F, Guzman-Parra J. The Impact of COVID-19 Confinement on Cognition and Mental Health and Technology Use Among Socially Vulnerable Older People: Retrospective Cohort Study. J Med Internet Res 2022;24(2):e30598 URL: https://www.jmir.org/2022/2/e30598. DOI: 10.2196/30598 (Page 5, line 117)

2. Despite being good, this article could be improved a bit. The reviewer's comments must be corrected on some important parts of the manuscript, there are no gaps in the research, and the research methods are available.

We thank the editor for the comments. We have addressed the comments below. 

a) Identification of the research gap

We have more clearly identified the research gap, by stating what is available in the current literature and what is missing, along with the information we desire to gain from conducting this research. We have inserted the following portion in the manuscript in page 6, line 135 to 143.

“This study aims to close two main research gaps. Other studies have examined the relationship between CT acceptance and competency on CT use [18], as well as CT use and emotional support and SWB. Much of this research focused on older adults without examining the mediating effect of living arrangements. Hence, this study aims to assess the association between CT acceptance (competency) and CT use specifically amongst older adults living alone. Additionally, there is little evidence in the current literature linking CT acceptance and competency with emotional support and SWB. Hence the study also aims to determine the association between CT acceptance (competency) and their roles in providing emotional support and SWB amongst older adults living alone.”

b) Research methods in the abstract section 

We have edited the abstract, particularly the research methods section to ensure its alignment with the edited portions of the manuscript. The following insert has been added to the abstract on Page 2, line 37 and 42. 

“Participants were surveyed from September to November 2020... PROCESS Model 1 was used to estimate the conditional effects of CT acceptance (competency) on emotional support for those living alone versus with others. Following which, PROCESS Model 7 was used to estimate the conditional indirect effects of CT acceptance (competency) on SWB through emotional support.”

Response to Reviewers’ Comments 

Reviewer #2: This article is good, but there are some notes for improvement to make it better. First, the abstract must be corrected, where the abstract contains research methods. Second, explain in more detail along with the reasons why it is important to research your research study. Third, describe the research gap, between the current state and the desired condition, Fourth, explain what are the benefits for further research. Fifth, add journals from similar previous studies. Passion for improvement. Goodluck.

Thank you for your comments. We hope we have responded to your comments to satisfactorily.

1a) Revised abstract to include research methods 

We have edited the abstract, particularly the research methods section to ensure its alignment with the edited portions of the manuscript. The following insert has been added to the abstract on Page 2, line 37 and 42. 

“Participants were surveyed from September to November 2020... PROCESS Model 1 was used to estimate the conditional effects of CT acceptance (competency) on emotional support for those living alone versus with others. Following which, PROCESS Model 7 was used to estimate the conditional indirect effects of CT acceptance (competency) on SWB through emotional support.”

1b) Importance of conducting the current research 

We have included additional reasons regarding the importance of our research study in the manuscript. The following inserts have been included on Page 6, line 143 and 147. 

“Both pieces of information will be important to provide support to community care organisations and policymakers on how to address resistance and inertia amongst older adults living alone beyond the pandemic period and their daily lives. The information will also be useful in designing interventions to encourage older adults to use CT.”

1c) Describe the research gap, between the current state and desired condition.

We have more clearly identified the research gap, by stating what is available in the current literature and what is missing, along with the information we desire to gain from conducting this research. We have inserted the following portion in the manuscript in page 6, line 135 to 143.

“This study aims to close two main research gaps. Other studies have examined the relationship between CT acceptance and competency on CT use [18], as well as CT use and emotional support and SWB. Much of this research focused on older adults without examining the mediating effect of living arrangements. Hence, this study aims to assess the association between CT acceptance (competency) and CT use specifically amongst older adults living alone. Additionally, there is little evidence in the current literature linking CT acceptance and competency with emotional support and SWB. Hence the study also aims to determine the association between CT acceptance (competency) and their roles in providing emotional support and SWB amongst older adults living alone.”

1d) What are the benefits for further research. 

The benefits for further research have been included in the manuscript, Page 19, line 379 to 401. 

“Developing an understanding of the types of CT and their frequency of use could provide information to develop targeted interventions to encourage specific types of CT amongst older adults to improve their SWB… However, future research should aim to identify reasons for lack of social engagement to inform policies that can benefit this group of older adults. Accurate identification of these reasons could aid the respective support groups in understanding and targeting their respective needs and increase their social engagement and support from the community.” 

1e) Addition of new literature 

In response to the reviewer’s comments, we have included a list of literature from similar studies. 

16. Rolandi, E., Vaccaro, R., Abbondanza, S., Casanova, G., Pettinato, L., Colombo, M., & Guaita, A. (2020). Loneliness and social engagement in older adults based in Lombardy during the COVID-19 lockdown: The long-term effects of a course on social networking sites use. International journal of environmental research and public health, 17(21), 7912. (Page 4, line 88)

Rolandi et al., (2020) did a telephone survey on older adults aged 81 to 85, who previously participanted in a study aimed at evaluating the impact of social networking sites (SNSs) on self-perceived loneliness and social engagement with family and friends. Results suggest that participants who were trained to use SNS reported reduced feelings of being left out and lighter reduction in social contacts during COVID-19, as compared to participants who were not trained to use SNS. 

17. Li, Y., Godai, K., Kido, M. et al. Cognitive decline and poor social relationship in older adults during COVID-19 pandemic: can information and communications technology (ICT) use helps?. BMC Geriatr 22, 375 (2022). https://doi.org/10.1186/s12877-022-03061-z (Page 4, line 89)

Li et al., (2022) found that amongst older adults who were 80 years and above, non-information and communication technology (ICT) use was associated with cognitive decline, moderated by social isolation. They believe that for older adults who were vulnerable to poor social relationships, ICT use is potentially an efficient medium. 

18. Juvonen, J., Schacter, H. L., & Lessard, L. M. (2021). Connecting electronically with friends to cope with isolation during COVID-19 pandemic. Journal of Social and Personal Relationships, 38(6), 1782-1799. (Page 4, line 92)

Juvonen et al., (2021) conducted a cross-sectional survey with a sample of 295 participants (aged 18 to 70). Participants of all ages reported greater satisfaction with video calls during the imposed isolation. Overall findings highlight the potential psychological benefits of connecting electronically with close others, and suggest that connecting with friends offers a way to cope with imposed isolation as long as individuals are satisfied with their exchanges. 

19. Foong, H. F., Lim, S. Y., Rokhani, F. Z., Bagat, M. F., Abdullah, S. F. Z., Hamid, T. A., & Ahmad, S. A. (2022). For better or for worse? A scoping review of the relationship between internet use and mental health in older adults during the COVID-19 pandemic. International Journal of Environmental Research and Public Health, 19(6), 3658. (Page 4, line 94)

Foong et al., (2022) is a scoping review of the relationship between internet use and mental health in older adults during the COVID-19 pandemic. Their results suggest that Internet use for communication purposes seems to be associated with better mental health in older adults during COVID-19 pandemic. 

26. Dura-Perez E, Goodman-Casanova J, Vega-Nuñez A, Guerrero-Pertiñez G, Varela-Moreno E, Garolera M, Quintana M, Cuesta-Vargas A, Barnestein-Fonseca P, Gómez Sánchez-Lafuente C, Mayoral-Cleries F, Guzman-Parra J. The Impact of COVID-19 Confinement on Cognition and Mental Health and Technology Use Among Socially Vulnerable Older People: Retrospective Cohort Study. J Med Internet Res 2022;24(2):e30598 URL: https://www.jmir.org/2022/2/e30598. DOI: 10.2196/30598 (Page 5, line 117)

Dura-perez et al., (2022) explored the impact of the COVID-19 outbreak on older adults’ mental well-being (e.g. quality of life, depression). The study also analyzed the association with technophilia (positive and enthusiasm towards technology) on the outcomes. Results suggest that older adults with higher technophilia were associated with better mental outcomes.

In addition, we would like to highlight that we have already cited literature that examined similar areas of research in the current version of the manuscript. In the introduction, the following papers with a short description of each paper’s content, are included: 

4. Fingerman KL, Ng YT, Zhang S, Britt K, Colera G, Birditt KS, et al. Living alone during COVID-19: Social contact and emotional well-being among older adults. J Gerontol B Psychol Sci Soc Sci. 2021;76(3):e116–e21. doi: doi:10.1093/geronb/gbaa200. (Page 3, Line 62)

Fingerman et al., (2021) examined how living alone was associated with daily social contact and emotional well being among older adults during the pandemic. Results suggest that older adults who live alone were less likely to see others to receive or provide help. In-person contact was associated with more positive emotions and less negative emotions for older adutls living alone, as compared to phone contact. 

6. Noguchi T, Hayashi T, Kubo Y, Tomiyama N, Ochi A, Hayashi H. Living Alone and Depressive Symptoms among Older Adults in the COVID-19 Pandemic: Role of Non–Face-to-Face Social Interactions. Journal of the American Medical Directors Association. 2023;24(1):17-21.e4. doi: https://doi.org/10.1016/j.jamda.2022.10.014. (Page 4, line 90)

Noguchi et al., (2023) examined the association of living alone with changes in depressive symptom status, and the moderating effect of non-face-to-face social interactions (e.g. phone calls, emails) among older adults during the COVID-19 pandemic. Results suggest a protective effect of weekly non-face-to-face social interactions for depressive symptom onset for older adults living alone. 

11. Chopik WJ. The benefits of social technology use among older adults are mediated by reduced loneliness. Cyberpsychology, Behavior and Social Networking. 2016;19(9). doi: 10.1089/cyber.2016.0151. (Page 3, line 76)

Chopik (2016) examined the assoiation between technology use for social reasons and psychological health amongst older adults. Results suggest that high levels of social technology use was associated with higher subjective well-being and fewer depressive symptoms. Results also suggest that technology has the potential to cultivate successful relationships among older adults. 

12. Sims T, Reed AE, Carr DC. Information and communication technology use is related to higher well-being among the oldest-old. J Gerontol B Psychol Sci Soc Sci. 2017;72(5):761–70. doi: doi:10.1093/geronb/gbw130. (Page 3, line 76)

Sims et al., (2017) examined the use of information and communiation technology (ICT) in relation to social goals and higher well-being amongst the oldest-old. Results suggest that ICT use predicted higher well-beig across outcomes, mediated by social motivations to connect with others). 

13. Holtzman S, DeClerck D, Turcotte K, Lisi D, Woodworth M. Emotional support during times of stress: Can text messaging compete with in-person interactions? Computers in Human Behavior. 2017;71:130-9. doi: https://doi.org/10.1016/j.chb.2017.01.043. (Page 3, line 78)

Holtzman et al., (2017) compared the impact of in-person support to text messaging support. Results sgugest that in-erson support resulted in higher levels of positive emotions than texting. However, when interacting with close friends, satisfaction for both modes of communication was equivalent. 

14. Li J, Zhou X. Internet use and Chinese older adults’ subjective well-being (SWB): The role of parent-child contact and relationship. Computers in Human Behavior. 2021;119:106725. doi: https://doi.org/10.1016/j.chb.2021.106725. (Page 4, line 83)

Li et al., (2021) examines whether the frequency of parent-child contact and parent-child relationship can medicate the relationship between Internet use and Chinese older people’s subjective well-being. Results suggest that Internet use was associated with older adults’ subjective well-being, mediated by parent-child contact and relationship. Findings further suggest that Internet use may empower older adults to maintain close intergenerational relationships contributing to their subjective well-being. 

15. Kotwal AA, Holt-Lunstad J, Newmark RL, Cenzer I, Smith AK, Covinsky KE, et al. Social Isolation and Loneliness Among San Francisco Bay Area Older Adults During the COVID-19 Shelter-in-Place Orders. J Am Geriatr Soc. 2021;69(1):20-9. Epub 20201009. doi: 10.1111/jgs.16865. PubMed PMID: 32965024; PubMed Central PMCID: PMCPMC7536935. (Page 4, line 86)

Kotwal et al., (2021) aimed to investigates experiences of social isolation and loneliness during shelter-in-place orders, as well as unmet health needs related to changes in social interactions. Results suggest that most participants communicated via telephone interactions. Results from open-ended feedback by participant further suggest that access to technology was central to their ability to cope with restrictions and maintain social connections. 

In the discussion section, the following papers are included: 

45. Yu RP, McCammon RJ, Ellison NB, Langa KM. The relationships that matter: social network site use and social wellbeing among older adults in the United States of America. Ageing and Society. 2015;36(9):1826-52. doi: 10.1017/s0144686x15000677. (Page 17, line 340)

Yu et al., (2015) studied how the use of social network sites (SNSs) was associated with social wellbeing outcomes. Results suggest that SNS use is positively associated with non-kin related social wellbeing outcomes, including perceived support and connectedness from friends. 

46. Wu H-Y, Chiou A-F. Social media usage, social support, intergenerational relationships, and depressive symptoms among older adults. Geriatric Nursing. 2020;41(5):615-21. doi: https://doi.org/10.1016/j.gerinurse.2020.03.016. (Page 16, line 340)

Wu & Chiou (2020) examined using a cross-sectional, correlational study design on social media usage, social support, intergenerational relationships and depressive symptoms. Their findings suggest that social media usage and social support were negatively associated with depressive symptoms. They recommend promoting the use of social media to increase social interactions amongst older adults. 

49. Hu Y, Qian Y. COVID-19, Inter-household Contact and Mental Well-Being Among Older Adults in the US and the UK. Frontiers in Sociology. 2021;6. doi: 10.3389/fsoc.2021.714626. (Page 17, line 348)

Hu & Qian (2021) examined how inter-household contact in face-to-face and virtual forms (separately and in combination) related to older adults’ mental well-being during the pandemic. Study findings suggest the importance of social communication (especially face-to-face contact) in sustaining older adults’ mental well-being.

 If the reviewer has any suggestions for relevant study, please feel free to recommend them to use for consideration.

---

## [Editor Report · Decision Letter 2]

6 Sep 2023

Acceptance of Communication Technology, Emotional Support and Subjective Well-being for Chinese Older Adults Living Alone during COVID-19: A Moderated Mediation Model

PONE-D-22-27277R2

Dear Dr. Nai,

We’re pleased to inform you that your manuscript has been judged scientifically suitable for publication and will be formally accepted for publication once it meets all outstanding technical requirements.

Kind regards,

Bing Han, M.D.

Academic Editor

PLOS ONE

---

## [Editor Report · Acceptance letter]

14 Sep 2023

PONE-D-22-27277R2 

Acceptance of Communication Technology, Emotional Support and Subjective Well-being for Chinese Older Adults Living Alone during COVID-19: A Moderated Mediation Model 

Dear Dr. Nai:

I'm pleased to inform you that your manuscript has been deemed suitable for publication in PLOS ONE. Congratulations! Your manuscript is now with our production department. 

Kind regards, 

on behalf of

Dr. Bing Han 

Academic Editor

PLOS ONE